# Strengths and Difficulties among Adolescent with and without Specific Learning Disorders (SLD)

**DOI:** 10.3390/children10111741

**Published:** 2023-10-27

**Authors:** Bettina F. Piko, Réka Dudok

**Affiliations:** 1Department of Behavioral Sciences, University of Szeged, 6722 Szeged, Hungary; 2Doctoral School of Education, University of Szeged, 6722 Szeged, Hungary; dudok.reka@edu.u-szeged.hu

**Keywords:** Specific Learning Disorder (SLD), emotional problems, peer problems, prosocial behavior, proactive coping, SES self-assessment, parental education

## Abstract

Specific Learning Disorders (SLD) have become a major concern in modern societies. It is essential to detect their emotional, behavioral and social consequences as early as childhood. The aim of this study is to examine a set of strengths and difficulties and compare them between students with and without SLD. Participants in this study were adolescents aged 11–18 years from Budapest and villages of its Metropolitan area (Hungary) (N = 276, mean age = 13.6 years, SD = 1.8, 54.7% boys). Due to multistage sampling, a nearly equal number of students had SLD or not. In addition to sociodemographics, the Strengths and Difficulties Questionnaire (SDQ), Satisfaction With Life Scale and the Proactive Coping Inventory were included in the survey, and *t*-test, correlation and logistic regression analysis were applied in statistical analyses. Our findings suggest that in early adolescence (ages 11–14 years), conduct and peer problems, in late adolescence (ages 15–18 years), emotional problems, highlighted SLD. In terms of strengths, prosocial behavior in children with SLD may compensate difficulties, especially at a younger age. Students from lower SES families and those having parents with a lower educational level are more likely to have a diagnosis of SLD. Teachers and special educators should take care of improving the adolescents’ prosociality, social and coping skills and listening to emotional, conduct and peer problems in those with SLD.

## 1. Introduction

Based on the OECD declaration, students with special educational needs are divided into three categories: students with disabilities or impairments viewed as organic disorders; students with behavioral or emotional disorders, or specific learning difficulties; and students with disadvantages arising primarily from socio-economic, cultural and/or linguistic factors [1]. Specific Learning Disorders (SLD) are one of the most significant and frequent disorders among school-aged children including the heterogeneous group of learning difficulties. The Fifth Edition of the Diagnostic and Statistical Manual of Mental Disorders (DSM 5) considers Specific Learning Disorders (SLD) as a type of neurodevelopmental disorder that worsens the arithmetic, writing or reading ability or other academic skills and decreases one’s learning capacity [2]. While those with a clinical diagnosis of SLD usually meet the criteria of educational difficulties, among those with learning disabilities identified by the school system, not everyone would receive a clinical diagnosis according to DSM 5. Although learning disorders may be detected as early as the preschool years, usually they are diagnosed in primary schools, while there is a significant increase in the number of students with a diagnosed SLD in higher education [3]. Learning difficulties influence not only children’s cognitive and academic functioning but also their psychological functioning and mental well-being throughout the lifespan [4]. For example, among adults with reading difficulties, a high proportion of them reported problems with everyday psychosocial functioning due to the social and emotional consequences of the disability [5]. Among university students with SLD, lower self-esteem but higher levels of somatic complaints and anxiety symptoms, more social problems and negative emotions, such as depression scores, were found [3,6].

It is now a worldwide tendency to apply the model of inclusive schooling, that is, all students should be included in the general-education classrooms [7]. When these students are given specially designed instructions to meet their needs, they have a better chance to obtain higher achievement in addition to skills development to improve socially and behaviorally [8]. Unfortunately, not every school is appropriate for providing an ideal learning environment for helping all students fulfill their potential, particularly those with special needs [9]. For example, despite the existence of regulations, guidelines and instructions in Hungary regarding the suitable format and caliber of education for individuals with learning disabilities, not all regions are able to provide the necessary level of integration in education and training. This is primarily due to the inadequate number of special education teachers and the old-fashioned teaching approaches employed by the aging teaching staff. Not surprisingly, these children often face difficulties in the classrooms and experience emotional and behavioral problems. For example, learning disorders have been associated with lower levels of self-esteem as well as with a number of mental health problems, such as anxiety and depression, attention deficits or conduct disorders [10].

Globally, it is estimated by the World Health Organization that 14% (1 in 7) among youth aged between 10 and19 years experience problems with mental health and well-being, and many of them remain unrecognized [11]. This is especially true in the case of children with special educational needs (SEN), among others, those with Specific Learning Disorders (SLD). Therefore, screening for mental health problems among them is essential to prevent later adult psychiatric disorders as well as improving general mental health.

In order to enhance well-being among adolescents with SLD, we should improve their self-concept and develop character strengths (e.g., social intelligence, perseverance, gratitude etc.), particularly since elimination of external factors are rather difficult or sometimes even impossible to change. These strengths can elevate the students’ life satisfaction and well-being and help prevent anxiety or other psychopathologies [12]. Thus, for developing effective preventive programs, we should concentrate on both strengths and difficulties.

In terms of difficulties, children with SLD have a high risk of reporting emotional, behavioral problems or difficulties in social situations: frequencies of behavioral and emotional problems among them may be as high as 30% [13]. Using the Strengths and Difficulties Questionnaire (SDQ) [14], 47.8% of children with SLD aged between 7 and 15 years had difficulties in a Turkish study [15]. When quality of life among schoolchildren is compared with their typically developing peers, problems (difficulties) may be detected not only in school functioning but also in their social relationships with peers (social domain) and emotional well-being, self-esteem and mental health (psychological domain) [16]. Comorbidities are also common in children with SLD. In a sample of German 9–11-year-old children, the following rates were found: depression (28%), ADHD (28%), conduct disorders (22%) and anxiety disorders (21%) [17]. Hyperactivity and conduct disorder can be partly explained by similarity in working memory deficits [18]. However, other forms of problem behavior may be a reaction to the stress stemming from poor academic achievement and school failures.

Among strengths, prosocial behavior (that is a tendency to benefit others, i.e., helping and sharing) is a less explored concept in relation to learning difficulties, while among autistic children it has been found that they tend to employ different strategies during some prosocial behaviors [19]. Likewise, lower levels of prosocial behavior were detected among children with intellectual disability as compared to their typically developing peers [20]. However, in a multiethnic study, adolescents with SLD showed better scores on the prosocial subscale of SDQ than those with global developmental delay or other developmental disorders [21]. In a Czech study, students with dyslexia did not differ from their peers in the levels of prosocial behavior [22]. It would be useful to detect prosocial behavior among children with SLD since it may contribute to their positive psychosocial development, enhance building peer relations and help create a comfortable classroom climate [23]. Enhancing social support in the classroom would be particularly necessary since students with SLD often report lower level of social support from their peers and teachers, and they experience fewer positive relationships with them and develop poorer attachment to school [24,25]. All these negative experiences may negatively contribute to their self-concept and academic identity [26].

In addition to prosocial behavior, another less investigated strength is whether these students use effective coping strategies, e.g., proactive coping. In a study from India, students with learning disabilities aged between 15 and 17 years were compared with those without SLD. Findings showed that students with SLD had lower levels of emotional and social adjustment and applied proactive coping strategies less often [27]. Proactive coping as a type of anticipatory and preventive coping strategies is focused on goal pursuit in terms of future challenges; thus, applying this coping strategy would be effective in affect regulation and prevention of psychosocial maladjustment in the classroom. Findings from an Italian study presented that several coping strategies were applied more common (e.g., aggressive coping or escaping), whole others were less used (e.g., active coping) [28].

In this study, we aim to examine a set of strengths and difficulties and compare them between students with and without SLD. In order to obtain a picture about strengths and difficulties as well as well-being among these students, we applied the Strengths and Difficulties Questionnaire (SDQ). This is a commonly used measure of child and adolescent psychosocial functioning consisting of 25 items in five domains: emotional symptoms, conduct problems, hyperactivity, peer problems and prosocial behavior [29]. In addition, we also included two psychological variables (strengths) which may contribute to improvement of students’ well-being, namely, satisfaction with life and proactive coping. Furthermore, we also controlled for sociodemographics (sex, age, father and mother schooling, SES self-assessment). Studies usually found that learning disabilities are still more common among children from lower socioeconomic households [15,30]. In addition, parents with lower levels of educational attainment are usually less engaged in their children’s education, i.e., home-based learning [31]. Age also seems relevant in understanding the nature of difficulties in early and late adolescence: while early teenage years are critical in terms of impulse control and development of self-esteem, adolescents gain more independence from their parents later with more autonomy in decision making, but more socioemotional and mental health problems, e.g., depression [32]. Particularly the transition from middle to high school brings about many new challenges and new types of problem behavior, such as internalizing behavior or substance use [33]. Thus, from the psychological and developmental point of view, a distinction between early (from puberty up to 14 years) and late adolescence (from 15 to 18 years) seems appropriate along the development of biological structures, emotional, social and cognitive skills [34]. Therefore, due to a wide age range of our sample, we investigated contributors to SLD in two age groups as well: in a younger age group (i.e., early adolescence, aged between 11 and 14 years) and an older age group (i.e., late adolescence, aged between 15 and 18 years). In terms of analyses, first we compared levels of these scales along the presence or absence of SLD; then we determined their contribution to differences in the odds of adolescents’ SLD by means of logistic regression analyses.

## 2. Materials and Methods

### 2.1. Participants and Procedure

A total of 276 adolescents (aged between 11 and 18 years, mean age = 13.57 years, SD = 1.81, 54.7% boys) participated in our survey who studied in primary and high schools from grades 5 to 12 in Budapest and its Metropolitan area (Hungary). Due to the purpose of our study (that is, to compare strengths and difficulties between adolescents with and without SLD), a multi-stage sampling has been applied to obtain a nearly equal number of students in each group. Thus, data collection was based on a judgmental sampling rather than a representative one. Using this design, in the sample, 51.8% (*n* = 143) had special educational needs due to Specific Learning Disorders. Of these, 95% of the sample (136 children) had a mixed disorder, with difficulties in all three areas (dyslexia, dysgraphia and dyscalculia), 49 of these had a diagnosis of F83—mixed specific developmental disorders and 87 had a diagnosis of F81.3—mixed disorder of scholastic skills according to the ICD-10 categorization. In the sample, 2 persons had dyscalculia only (diagnosed with F81.2 according to ICD-10), 3 persons had dysgraphia only (diagnosed with F81.1 according to ICD-10) and 2 persons had dyslexia and dysgraphia combined (diagnosed with F81.0 and F81.1 according to ICD-10). One case of ADHD and one case of autism were comorbid diagnoses. Due to the relatively low sample size and various diagnoses, subgroups could not be further defined in the analyses. The completion of the questionnaire was anonymous and voluntary. Prior to data collection, parental informed consent was obtained. The ethical approval was provided by the Ethics Committee of the Doctoral School of Education, University of Szeged, Hungary. The data collection was going on during the school year of 2021/2022 by means of a self-administered questionnaire in paper–pencil form. A special educator from the school helped the students with administration and described a diagnosis on the form; otherwise, the completion was anonymous. During the recording, those who had problems with reading or interpreting the questions were read out by these facilitators. However, we did not group the children, only the questionnaires. The questionnaire could be completed within approximately 20 min. With only five children declining participation in the survey, the response rate was 98%. A pilot study with selected children from different age groups justified that their cognitive ability and load was acceptable and did not mean a bias affecting the results. Questionnaires with too many incomplete responses (only a few of them), were skipped from data analysis.

### 2.2. Measurements

Specific Learning Disorder as a diagnosis was described on the questionnaire by a special educator to obtain an objective description. In addition to sociodemographics (age, sex, SES father and mother schooling, self-assessment), the questionnaire contained measurements on a number of strengths and difficulties, based on various scales. SES self-assessment was a subjective evaluation of the family’s socioeconomic status in a 5-point rating scale: upper, upper–middle, middle, lower–middle or lower class [35]. In support of our aim for the current analysis, this variable was dichotomized (upper and upper middle class vs. middle or below). Originally, father and mother schooling included five levels: primary education, apprenticeship, secondary modern school, high school/grammar school and university or college degree. Likewise, for the variable SES self-assessment, educational level was also recoded into two groups: “without General Certificate of Education” (primary education, apprenticeship) and those “with GCE” (secondary modern school, high school and college/university).

The adolescents were asked to complete the Hungarian adapted and validated version [36] of the Strength and Difficulties Questionnaire (SDQ) for 11–17 year olds [29]. The self-rated scale has been widely utilized to screen for mental and behavioral health problems in children and adolescents in the general population. It can be downloaded from the official website of the SDQ (https://www.sdqinfo.org/, accessed on 20 January 2023). The scale included 25 core items on specific strengths and difficulties with an overall rating of whether the children have emotional or behavioral problems. Each of these items is rated as being not true (0), somewhat true (1) or certainly true (2). The subscales consist of five items, thus yielding scores between 0 and 10. Among the five subscales, four refer to difficulties (emotional problems, conduct problems, hyperactivity and peer problems), while the fifth refers to strength (prosocial behavior). Reliability coefficients (Cronbach alphas) varied between α = 0.69 (Hyperactivity) and α = 0.77 (Prosocial behavior).

The Proactive Coping Scale is part of the Proactive Coping Inventory, and it was developed to measure problem-focused coping strategies which may play a role in adaptation in difficult situations by proactively assessing future goals to achieve them successfully [37]. The scale consists of 14 items which summarize responses and reactions to specific life situations and ask respondents to indicate on a 4-point Likert scale how they experience the situation (e.g., “When I experience a problem, I take the initiative in resolving it.”). The Hungarian validated version was applied in this study [38]. Higher scores indicate a greater tendency to use this type of coping. The scale was reliable with a Cronbach’s alpha = 0.83 in this study.

Finally, the Hungarian validated version [39] of The Satisfaction With Life Scale (SWLS) [40] was used to assess the global level of life satisfaction, as a measurement of general subjective well-being. The respondents should indicate how strongly they agreed with each of the five items (e.g., “The conditions of my life are excellent.”). Responses varied from strongly disagree (1) to strongly agree (7). The final scale had a range of 5–35, where the higher score indicated a greater level of life satisfaction. The scale was reliable, with a Cronbach’s alpha of 0.81 with the current sample.

### 2.3. Data Analysis

SPSS for MS Windows Release 25.0 program was used in the calculations and the maximum significance level set to 5%. First, descriptive statistics were used to detect group differences in the scores for study scales by sex and SLD status. Student *t*-tests were applied to approve statistical significance. Second, correlation coefficients between the scales were also detected. Subsequently, we calculated bivariate (binary) logistic regression analyses at the 95% probability level to determine the effect of each independent variable on increasing or reducing the odds of SLD diagnosis including sociodemographics (age, sex, SES self-assessment and parents’ schooling), scores of SDQ subscales, proactive coping and life satisfaction. Finally, multivariate logistic regression was applied to determine the most relevant predictors. A positive association (i.e., an elevated likelihood) between the outcome variable (presence of SLD) and other study variables is based on an odds ratio (OR) > 1.0, while a value < 1.0 indicates a lower likelihood. Statistical significance was defined by two criteria: a maximum *p*-value of 0.05, and a 95% confidence interval (CI) does not include 1.0.

## 3. Results

### Descriptive Statistics

In terms of the sociodemographic characteristics of the children (see Table 1), in the sample there were 151 boys (54.7%) and 125 girls (45.3%); 154 (55.8%) were between the ages of 11 and 14 years and 122 (44.2%) aged between 15 and 18 years. Father schooling categories were the following: 2.7% with primary education, 23.4% with apprenticeship, 35.3% with secondary modern school, 12.7% with high school/grammar school and 15.9% with university or college degree. Mother schooling categories were the following: 12.2% with primary education, 21.4% with apprenticeship, 24.8% with secondary modern school, 13.0% with high school/grammar school and 28.6% with university or college degree. In dichotomous form: 33% of fathers and 31.9% of mothers did not have a General Certificate of Education (equivalent with a graduation from high school), while other parents had higher levels of education. In the sample, 6.5% were evaluated as belonging to the upper class and 37.3% to upper–middle class; 48.2% to middle class and the remaining students noted lower–middle (6.5%) and lower class (1.4%). In subsequent analyses we use this variable in dichotomous form (middle class or below: 56.2%; upper and upper–middle class: 43.8%). Of the students, 143 (51.8%) had a diagnosis of SLD.

Table 2 presents descriptive statistics for the study scales by sex. Girls scored significantly higher on the “Emotional problems” subscale (t(260) = −5.18, *p* < 0.001) and on the “Prosocial behavior” subscale (t(265) = −2.31, *p* < 0.05). On the other hand, they scored significantly lower on the scale of life satisfaction (t(266) = 1.97, *p* < 0.05).

Table 3 shows descriptive statistics for the study scales by SLD status. Those with a SLD diagnosis reported significantly more emotional problems (t(260) = −3.01, *p* < 0.01) and peer problems (t(264) = −3.37, *p* < 0.001). Whereas these adolescents scored lower on the life satisfaction scale (t(266) = 2.11, *p* < 0.05) and used less proactive coping strategies (t(256) = 2.35, *p* < 0.05).

Table 4 presents Pearson correlation coefficients for the study scales. The Emotional problems variable was positively correlated with conduct problems (r(260) = 0.29, *p* < 0.001), hyperactivity (r(261) = 0.29, *p* < 0.001), peer problems (r(260) = 0.34, *p* < 0.001) and negatively with life satisfaction (r(256) = −0.34, *p* < 0.001) and proactive coping (r (250) = −0.22, *p* < 0.01). There was a strong positive association between conduct problems and hyperactivity (r(266) = 0.54, *p* < 0.001). While the variable “Conduct problems” was positively related to peer problems (r(264) = 0.30, *p* < 0.001), there was no association between peer problems and hyperactivity. Both conduct problems and hyperactivity were negatively correlated with prosocial behavior, life satisfaction and proactive coping. Prosocial behavior showed a negative association with peer problems (r(263) = −0.15, *p* < 0.05) and a positive correlation with life satisfaction (r(262) = 0.31, *p* < 0.001) and proactive coping (r(254) = 0.36, *p* < 0.001). Finally, satisfaction with life was positively associated with proactive coping (r(252) = 0.44, *p* < 0.001). Age was only correlated with life satisfaction (r(268) = −0.15, *p* < 0.05).

Table 5 displays the results of simple binary logistic regression analyses (odds ratios and 95% confidence intervals) for a diagnosis of SLD.

Among sociodemographics, age (OR = 1.15; 95% CI = 1.01–1.31, *p* < 0.05) increased, whereas higher levels of father schooling (OR = 0.42; 95% CI = 0.25–0.71, *p* < 0.001) as well of mother schooling (OR = 0.57; 95% CI = 0.34–0.94, *p* < 0.05) decreased the likelihood of having SLD as a diagnosis. A similar association was detected in the case of higher levels of SES self-assessment (OR = 0.39; 95% CI = 0.24–0.64, *p* < 0.001): it decreased the risk of having a Specific Learning Disorder. Among the SDQ subscales, both emotional (OR = 1.16; 95% CI = 1.28–0.94, *p* < 0.001) and peer problems (OR = 1.22; 95% CI = 1.08–1.38, *p* < 0.001) were significantly correlated with a SDL diagnosis. Finally, satisfaction with life (OR = 0.96; 95% CI = 0.93–0.99, *p* < 0.05) and proactive coping (OR = 0.96; 95% CI = 0.92–0.99, *p* < 0.05) showed a lower likelihood of SLD (Table 5).

Table 6 displays the results of simple binary logistic regression analyses in two age groups: for 11–14-year olds and 15–18-years olds. In both groups, the total SDQ score predicted the presence of SLD diagnosis. However, in the younger age group, all socioeconomic indicators were significant predictors. While among the difficulties, conduct and peer problems served as contributors to SLD in younger age, among the older adolescents, emotional problems seemed to play a similar role. Among the strengths, proactive coping was significantly related to SLD only in the younger age group.

Finally, Table 7 and Table 8 present the results of the multivariate logistic regression analysis. All of the sociodemographics variables proved significant predictors with an exception of mother schooling: age (OR = 1.18; 95% CI = 1.01–1.40, *p* < 0.05), being a female (OR = 0.40; 95% CI = 0.18–0.71, *p* < 0.01), having a SES background with upper/upper-middle class (OR = 0.45; 95% CI = 0.24–0.83, *p* < 0.05) and having a father with General Certificate of Education or more (OR = 0.36; 95% CI = 0.17–0.78, *p* < 0.01).

Among the psychological scales representing strength and difficulties, the variable named “Emotional problems” remained a significant predictor (OR = 1.15; 95% CI = 1.01–1.32, *p* < 0.05). In addition, prosocial behavior proved to be a predictor in the multivariate analysis (OR = 1.31; 95% CI = 1.10–1.56, *p* < 0.01) (Table 7).

While in the younger age group, two strengths remained significant (father schooling: OR = 0.23; 95% CI = 0.09–0.58, *p* < 0.01 and prosocial behavior: OR = 1.37; 95% CI = 1.11–1.69, *p* < 0.01), among older adolescents, being a female (OR = 0.16; 95% CI = 0.05–0.54, *p* < 0.01) was a protective factor, in addition to emotional problems remained a significant difficulty (OR = 1.39; 95% CI = 1.12–1.73, *p* < 0.01). The goodness of fit was significant in all cases (Table 8).

## 4. Discussion

While Specific Learning Disorders (SLD) have become a major concern in educational settings and health care in modern societies worldwide [41], it is essential to identify their emotional, behavioral and social consequences to minimize them for the individuals, their families and whole societies. Searching for strengths which may act as protective factors may be even more relevant in terms of prevention. In addition to identifying the psychiatric comorbid states, such as ADHD, depression or anxiety disorders, concentrating on social and coping skills may help prevent deterioration of mental health and well-being of students with SLD. In this study, we aimed to compare students with and without SLD along the lines of difficulties and strengths. Our results support previous findings that emotional problems are indeed a common experience the students with SDL should face [10,16,17], especially in the older age group. On the contrary, in the younger age group, peer and conduct problems were the most relevant difficulties. In addition, children with SLD were less satisfied with their life and tended to apply proactive coping strategies less often, whereas they showed more prosociality. Among the strengths, prosocial activity seemed to play a beneficial role in children’s lives with SLD, particularly in younger age. Finally, as expected, social inequalities appeared in the occurrence of SLD.

First of all, the most relevant finding is that students with SLD should face more difficulties in social, emotional and behavioral domains compared to their typically developing peers, similar to previous studies [10,16,17]. However, there are differences in the nature of problems in various age groups. While younger adolescents should face conduct and peer problems, older adolescents experience a challenge of emotional difficulties. This difference may stem from age characteristics, e.g., conduct and behavioral problems usually start at the age of 11 or before [42]. Late adolescents, on the other hand, usually show more emotionality and are more prone to engage in substance use [32,43]. Since western societies are success oriented and celebrate cognitive capacity, it is a great burden on children and the families to cope with negative attitudes toward failures [5]. Academic failures may have a negative impact on the children’s self-concept and self-esteem [26], which can deteriorate their peer relationships [16,24,25]. During adolescence, the children’s social network undergoes important changes, and peer connections become especially decisive in their psychosocial development [32,44]. In contrast with this experience, the level of hyperactivity among these students was only slightly higher which did not reach statistical significance.

Another relevant finding was the role of prosocial behavior. Although average score of the prosocial subscale was only slightly higher in students with SDL, in multivariate analysis it proved a significant predictor, especially among 11–14-year olds. In contrast with previous results [21,22], students with SLD showed higher levels of prosocial behavior compared to their typically developing peers. A recent study also reported a higher level of prosociality among them compared to children with global developmental delay or other developmental disorders [20]. We assume that these adolescents may make an effort to build supportive relationships to avoid loneliness and peer problems by helping others and showing positive attitudes toward their peers, particularly those in trouble (see items such as “I try to be nice to other people” or “I am helpful if someone is hurt, upset or feeling ill”). This finding suggests that negative experiences may help develop sympathy toward others. It is an important strength for them since prosocial behavior shows a positive correlation with life satisfaction and proactive coping, and negative association with conduct and peer problems and hyperactivity. Unlike these problems, the relationship with emotional problems seems nonsignificant. We hypothesize that those with more emotional problems can show less prosocial behavior due to the students’ own negative feelings. On the other hand, positive feelings may reward and promote prosocial behavior [45]. Since previous studies also draw our attention to the positive psychological consequences of prosocial behavior, it would be useful to conduct further studies and interventions on promoting prosocial behavior, including both students with and without SLD.

In addition to prosocial behavior, proactive coping seems an effective strength in psychosocial development since it may reduce levels of stress with active problem resolution [34]. In line with this concept, we found that proactive coping was negatively correlated with all difficulties, namely, emotional, conduct and peer problems and hyperactivity, and positively with life satisfaction and prosocial behavior. Previous studies reported lower tendencies of applying this coping strategy among children with SLD and SEN [27,28]. Although this variable did not remain a significant predictor in multivariate analysis, proactive coping scores were significantly lower among adolescents with SLD compared to their typically developing peers.

Finally, we should also mention sociodemographics. In epidemiological studies, more males are diagnosed with dyslexia than females [46], and our study also supports that being a female may decrease the risk of SLD. Although the level of emotional problems was higher among girls, this was not the case with other difficulties. Lower socioeconomic status and lower levels of parental—particularly father’s—education may also act as a risk factor similar to previous studies [30,31]. In terms of dyslexia, home literacy environment may have a great influence on children’s reading development [47], and parental education mediates this relationship. It may also happen that parents of these children had similar learning difficulties; however, we do not have enough information on it.

Among the strengths of our study, we should mention the nearly equal number of adolescents (i.e., matched samples) with and without SLD, and applying a validated and previously adapted collection of scales and measurements. We highlighted some relevant new findings on the role of strengths which have been less investigated in relation to SLD, such as prosocial behavior or proactive coping. We also put a special emphasis on the families’ social background. We should also mention here that studies on children with SLD is a less investigated field of research in Hungary, therefore, these findings represent a unique contribution to our knowledge of their problems. However, we should also mention some limitations. Due to the cross-sectional study design, we cannot interpret causality in relationships. In addition, the specific sampling may limit the generalizability of the results. Since our sample size was relatively small, it did not make possible to analyze these relationships in different groups based on categories of SLD; a larger study would be more appropriate for future investigations. Using a self-report questionnaire, it might happen that children tend to be less aware of their weaknesses and strengths in self-reports. Finally, the reliability values were lower, as we had expected in terms of the SDQ.

## 5. Conclusions

We can conclude that findings of this study highlight some relevant difficulties and strengths in a sample of adolescents. These findings indicate that several variables differentiated adolescents with and without SLD: emotional, conduct and peer problems, prosocial behavior, proactive coping and life satisfaction. In addition to sociodemographics (sex, age, SES self-assessment and parental schooling) being the most decisive predictors of SLD, different difficulties and strengths may play a role in children’s lives with a diagnosis of SLD. In addition, an important difference may arise from the role of age: at younger age, conduct and peer problems, in late adolescence, emotional problems are the most relevant difficulties. Overall, our findings suggest the following: (a) in early adolescence, conduct and peer problems, in late adolescence, emotional problems were the most relevant difficulties highlighting SLD; (b) prosocial behavior in children with SLD may compensate difficulties, especially in younger children; and (c) students from lower SES households and those having parents with lower education are more likely to have a diagnosis of Specific Learning Disorder with special educational needs in school setting.

An important message of these results is that children with SLD may have higher chance to experience emotional, conduct and peer problems in the classrooms and educators should take care of them to prevent social conflicts and social stigma. Prosocial behavior can act as a social asset to positively contribute to social relationship with peers, and while children without SDL tend to use more prosocial behaviors, every student needs to improve social skills and prosociality to build supportive social relationships with each other. Finally, those from lower social classes and having parents with lower schooling need special attention and more help with the development of reading and writing skills and literacy.

## Figures and Tables

**Table 1 children-10-01741-t001:** Descriptive statistics for sociodemographics and sample characteristics (N = 276).

Variables	*n* (%)
**Sex**	
Boys	151 (54.7)
Girls	125 (45.3)
**Age group**	
11–14 years	154 (55.8)
15–18 years	122 (44.2)
**SES self-assessment**	
Middle class or lower	155 (56.2)
Upper and Upper-middle class	121 (43.8)
**Father schooling**	
Less than GCE	91 (33.0)
GCE or more	185 (67.0)
**Mother schooling**	
Less than GCE	88 (31.9)
GCE or more	185 (68.1)
**Specific Learning Disorder (SLD)**	
No	133 (48.2)
Yes	143 (51.8)

Notes: GCE: General Certificate of Education.

**Table 2 children-10-01741-t002:** Descriptive statistics for study scales by sex (N = 276).

Variables	Male (*n* = 151)	Female (*n* = 125)	*t*-Value
	Mean ± SD	Mean ± SD	
**SDQ dimensions**			
Emotional problems	3.17 ± 2.32	4.76 (2.65)	**−5.18 ****
Conduct problems	3.03 ± 1.94	2.83 ± 1.76	0.90
Hyperactivity	4.22 ± 2.14	4.17 ± 2.31	0.17
Peer problems	2.96 ± 2.09	2.71 ± 2.12	0.96
Prosocial behavior	7.19 ± 2.11	7.76 ± 1.93	**−2.31 ***
**Subjective well-being**			
Satisfaction with Life	25.34 ± 6.84	23.67 ± 7.06	**1.97 ***
**Coping skill**			
Proactive coping	42.03 ± 6.99	39.24 ± 6.58	3.28

Notes: Student *t*-tests: * *p* < 0.05; ** *p* < 0.001.

**Table 3 children-10-01741-t003:** Descriptive statistics for study scales by groups with or without a diagnosis of SLD (N = 276).

Variables	Without a Diagnosis of SLD (*n* = 133)	With a Diagnosis of SLD (*n* = 143)	*t*-Value
	Mean ± SD	Mean ± SD	
**SDQ dimensions**			
Emotional problems	3.40 ± 2.60	4.35 ± 2.51	**−3.01 ****
Conduct problems	2.75 ± 1.76	3.12 ± 1.94	−1.66
Hyperactivity	3.98 ± 2.11	4.40 ± 2.30	−1.51
Peer problems	2.41 ± 2.13	3.26 ± 2.00	**−3.37 *****
Prosocial behavior	7.21 ± 2.03	7.67 ± 2.05	−1.83
**Subjective well-being**			
Satisfaction with Life	25.52 ± 6.35	23.73 ± 7.44	**2.11 ***
**Coping skill**			
Proactive coping	41.78 ± 6.68	39.77 ± 7.05	**2.35 ***

Notes: Student *t*-tests: * *p* < 0.05; ** *p* < 0.01; *** *p* < 0.001.

**Table 4 children-10-01741-t004:** Correlation matrix for the study scales.

Variables	1	2	3	4	5	6	7
1. Emotional problems	-	-	-	-	-	-	
2. Conduct problems	0.29 ***	-	-	-	-	-	
3. Hyperactivity	0.29 ***	0.54 ***	-	-	-	-	
4. Peer problems	0.34 ***	0.30 ***	0.12	-	-	-	
5. Prosocial behavior	0.07	−0.36 ***	−0.28 ***	−0.15 *	-	-	
6. Satisfaction with Life	−0.24 ***	−0.33 ***	−0.26 ***	−0.31 ***	0.31 ***	-	
7. Proactive coping	−0.22 **	−0.18 **	−0.30 ***	−0.22 **	0.36 ***	0.44 ***	
8. Age	0.08	−0.06	0.08	0.06	−0.01	−0.15 *	0.01

Notes: Pearson correlation coefficients: * *p* < 0.05; ** *p* < 0.01; *** *p* < 0.001.

**Table 5 children-10-01741-t005:** Bivariate logistic regression analysis of adolescents’ SLD diagnosis.

Predictors	Diagnosis of SLD	
	OR (95% CI)	B (S.E.) Wald
**Socio-demographic variables**		
Age (years)	**1.15 (1.01–1.31) ***	0.14 (0.07) 4.08
Sex		
Male ^a^	1.00	
Female	0.67 (0.42–1.08)	0.40 (0.24) 2.67
SES self-assessment		
Middle class or lower ^a^	1.00	
Upper and Upper-middle class	**0.39 (0.24–0.64) ****	−0.94 (0.25) 14.24
Father schooling		
Less than GCE ^a^	1.00	
GCE or more	**0.42 (0.25–0.71) ****	−0.87 (0.27) 10.61
Mother schooling		
Less than GCE ^a^	1.00	
GCE or more	**0.57 (0.34–0.95) ***	−0.57 (0.26) 4.68
**SDQ dimensions**		
Emotional problems	**1.16 (1.05–1.28) ****	0.15 (0.05) 8.57
Conduct problems	1.12 (0.98–1.27)	0.11 (0.07) 2.73
Hyperactivity	1.09 (0.97–1.21)	0.08 (0.06) 2.27
Peer problems	**1.22 (1.08–1.38) ****	0.20 (0.06) 10.54
Prosocial behavior	1.12 (0.99–1.26)	0.11 (0.06) 3.31
**Subjective well-being**		
Satisfaction with Life	**0.96 (0.93–0.99) ***	−0.04 (0.02) 4.37
**Coping**		
Proactive coping	**0.96 (0.92–0.99) ***	−0.04 (0.02) 5.33

Notes: SLD: Specific Learning Disorder. GCE: General Certificate of Education. SDQ: Strength and Difficulties Questionnaire. ^a^ Reference category; OR: odds ratio; CI: confidence interval; B: regression coefficient; S.E.: Standard Error. * *p* < 0.05; ** *p* < 0.001.

**Table 6 children-10-01741-t006:** Bivariate logistic regression analysis of adolescents’ SLD diagnosis in two age groups.

	Diagnosis of SLD Ages 11–14 Years		Diagnosis of SLD Ages 15–18 Years	
Predictors	OR (95% CI)	B (S.E.) Wald	OR (95% CI)	B (S.E.) Wald
**Socio-demographic variables**				
Sex				
Male ^a^	1.00		1.00	
Female	0.59 (0.34–1.03)	−0.53 (0.28) 3.45	0.80 (0.29–2.22)	−0.23 (0.52) 0.19
SES self-assessment				
Middle class or lower ^a^	1.00		1.00	
Upper and Upper-middle class	**0.35 (0.20–0.61) *****	−1.06 (0.29) 13.55	0.79 (0.27–2.31)	−0.23 (0.59) 0.19
Father schooling				
Less than GCE ^a^	1.00		1.00	
GCE or more	**0.29 (0.16–0.54) *****	−1.23 (0.31) 15.57	0.72 (0.25–2.07)	−0.32 (0.54) 0.37
Mother schooling				
Less than GCE ^a^	1.00		1.00	
GCE or more	**0.51 (0.29–0.92) ***	−0.67 (0.30) 5.02	0.65 (0.20–2.12)	−0.43 (0.60) 0.51
**SDQ dimensions**				
Emotional problems	1.12 (1.00–1.26)	0.11 (0.06) 3.80	**1.26 (1.03–1.56) ***	0.23 (0.11) 4.94
Conduct problems	**1.17 (1.07–1.35) ***	0.15 (0.07) 4.24	0.96 (0.70–1.31)	−0.04 (0.16) 0.06
Hyperactivity	1.08 (0.96–1.22)	0.08 (0.06) 1.69	1.05 (0.80–1.37)	0.05 (0.14) 0.12
Peer problems	**1.22 (1.07–1.40) ****	0.20 (0.07) 8.00	1.21 (0.92–1.60)	0.19 (0.14) 1.90
Prosocial behavior	1.13 (0.99–1.30)	0.12 (0.07) 3.21	0.99 (0.73–1.53)	−0.01 (0.15) 0.01
SDQ difficulties total	**1.08 (1.03–1.14) ****	0.08 (0.03) 9.54	**1.11 (1.01–1.22) ***	0.10 (0.03) 4.13
**Subjective well-being**				
Satisfaction with Life	0.97 (0.93–1.01)	−0.03 (0.02) 1.93	0.95 (0.88–1.03)	−0.05 (0.04) 1.35
**Coping**				
Proactive coping	**0.96 (0.92–0.99) ***	−0.04 (0.02) 3.97	0.95 (0.88–1.03)	−0.05 (0.04) 1.66

Notes: SLD: Specific Learning Disorder. GCE: General Certificate of Education. SDQ: Strength and Difficulties Questionnaire. ^a^ Reference category; OR: odds ratio; CI: confidence interval; B: regression coefficient; S.E.: Standard Error. * *p* < 0.05; ** *p* < 0.01; *** *p* < 0.001.

**Table 7 children-10-01741-t007:** Multivariate logistic regression analysis of adolescents’ SLD diagnosis.

	Diagnosis of SLD	
Predictors	OR (95% CI)	B (S.E.) Wald
**Socio-demographic variables**		
Age (years)	**1.18 (1.01–1.40) ***	0.17 (0.09) 3.83
Sex		
Male ^a^	1.00	
Female	**0.40 (0.18–0.71) ****	−1.02 (0.35) 8.57
SES self-assessment		
Middle class or lower ^a^	1.00	
Upper and Upper-middle class	0.45 (0.24–0.83) *	−0.80 (0.32) 6.46
Father schooling		
Less than GCE ^a^	1.00	
General Certificate of Education or more	**0.36 (0.17–0.78) ****	−1.02 (0.40) 6.68
Mother schooling		
Less than GCE ^a^	1.00	
General Certificate of Education or more	0.86 (0.40–1.87)	−0.15 (0.40) 0.14
**SDQ dimensions**		
Emotional problems	**1.15 (1.01–1.32) ***	0.14 (0.07) 3.90
Conduct problems	1.08 (0.86 –1.34)	0.07 (0.11) 0.44
Hyperactivity	0.99 (0.83–1.17)	−0.01 (0.09) 0.02
Peer problems	1.14 (0.97–1.95)	0.13 (0.08) 2.54
Prosocial behavior	**1.31 (1.10–1.56) ****	0.27 (0.09) 9.11
**Subjective well-being**		
Satisfaction with Life	0.98 (0.93–1.04)	−0.02 (0.03) 0.42
**Coping**		
Proactive coping	0.95 (0.90–1.00)	−0.05 (0.03) 3.86
χ^2^	63.29 ***	
df	12	
Nagelkerke R^2^	0.31	

Notes: SLD: Specific Learning Disorder. GCE: General Certificate of Education. SDQ: Strength and Difficulties Questionnaires. ^a^ Reference category; OR: odds ratio; CI: confidence interval; B: regression coefficient; S.E.: Standard Error; χ^2^: Chi-square; df: degree of freedom. * *p* < 0.05; ** *p* < 0.01; *** *p* < 0.001.

**Table 8 children-10-01741-t008:** Multivariate logistic regression analysis of adolescents’ SLD diagnosis in two age groups.

	Diagnosis of SLD Ages 11–14 Years		Diagnosis of SLD Ages 15–18 Years	
Predictors	OR (95% CI)	B (S.E.) Wald	OR (95% CI)	B (S.E.) Wald
**Socio-demographic** **variables**				
Sex				
Male ^a^	1.00		1.00	
Female	0.44 (0.20–0.96)	−0.82 (0.40) 4.28	**0.16 (0.05–0.54) ***	−1.80 (0.60) 8.95
SES self-assessment				
Middle class or lower ^a^	1.00		1.00	
Upper and Upper-middle class	0.49 (0.24–1.00)	−0.72 (0.37) 3.74	0.60 (0.24–1.53)	−0.50 (0.47) 1.13
Father schooling				
Less than GCE ^a^	1.00		1.00	
GCE or more	**0.23 (0.09–0.58) ***	−1.45 (0.46) 9.68	0.39 (0.12–1.30)	−0.94 (0.61) 2.32
Mother schooling				
Less than GCE ^a^	1.00		1.00	
GCE or more	1.24 (0.50–3.09)	0.22 (0.46) 0.23	1.15 (0.36–3.74)	0.15 (0.60) 0.06
**SDQ dimensions**				
Emotional problems	1.04 (0.88–1.23)	0.04 (0.08) 0.22	**1.39 (1.12–1.73) ***	0.33 (0.11) 8.64
Conduct problems	1.12 (0.87–1.46)	0.12 (0.13) 0.78	0.92 (0.66–1.28)	−0.08 (0.17) 0.23
Hyperactivity	1.02 (0.84–1.23)	0.02 (0.10) 0.04	0.93 (0.70–1.22)	−0.07 (0.14) 0.27
Peer problems	1.19 (0.98–1.45)	0.17 (0.10) 2.99	1.15 (0.89–1.48)	0.14 (0.13) 1.21
Prosocial behavior	**1.37 (1.11–1.69) ***	0.31 (0.11) 8.50	1.10 (0.85–1.43)	0.10 (0.13) 0.56
**Subjective well-being**				
Satisfaction with Life	0.98 (0.91–1.05)	−0.02 (0.03) 0.26	0.96 (0.89–1.04)	−0.04 (0.04) 1.13
**Coping**				
Proactive coping	0.95 (0.89–1.00)	−0.05 (0.03) 2.82	0.98 (0.91–1.06)	−0.02 (0.04) 0.31
χ^2^	48.75 **		26.27 *	
df	11		11	
Nagelkerke R^2^	0.32		0.29	

Notes: SLD: Specific Learning Disorder. GCE: General Certificate of Education. SDQ: Strength and Difficulties Questionnaires. ^a^ Reference category; OR: odds ratio; CI: confidence interval; B: regression coefficient; S.E.: Standard Error; χ^2^: Chi-square; df: degree of freedom. * *p* < 0.01; ** *p* < 0.001.

## Data Availability

The datasets generated during and/or analyzed during the current study are available from the corresponding author on reasonable request.

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
