# Peer review of "Strengths and Difficulties among Adolescent with and without Specific Learning Disorders (SLD)"

_children, 2023, doi:10.3390/children10111741_

Round 1
Reviewer 1 Report
Comments and Suggestions for Authors
Clearly the topic dealt with in this article is timely and relevant for Children. The paper is generally well-structured and well-written; however, we feel that the authors should address some issues:
(a) The literature in the introduction is very generic. It would be very enriching if they could expand the studies on SLD that have used the same SDG instrument, even if they have used parent or teacher estimates.
b) The description of the sample is very generic. A table with the sociodemographic information of the participants and their families should be included. In addition, the information that appears in the first paragraph of the results section (page 5) can easily be presented in this table.
c) It should be added as a limitation that a self-report questionnaire is used and that SLDs tend to be less aware of their weaknesses and strengths in self-reports.
Author Response
Dear Reviewer;
Thank you for your helpful and useful comments to improve the quality of our paper.
We revised our paper according to these comments. Please see below the changes we have made – in the text all corrections/additions are highlighted in red to make it easy to see what specific changes were made.
Reviewer 1
Clearly the topic dealt with in this article is timely and relevant for Children. The paper is generally well-structured and well-written; however, we feel that the authors should address some issues:
(a) The literature in the introduction is very generic. It would be very enriching if they could expand the studies on SLD that have used the same SDG instrument, even if they have used parent or teacher estimates.
RE: Thank you for this note. Based on your and the other Reviewer’s suggestions, we restructured the Introduction as follows: 1) recent concept and definitions; 2) difficulties among children with SLD; 3) Strengths among children with SLD. In addition, we also mentioned when SDQ was used in a concrete study. Some new references were also added. We really hope that Introduction is now better organized.
b) The description of the sample is very generic. A table with the sociodemographic information of the participants and their families should be included. In addition, the information that appears in the first paragraph of the results section (page 5) can easily be presented in this table.
RE: Thank you for this suggestion, we added a table with characteristics of the sample.
c) It should be added as a limitation that a self-report questionnaire is used and that SLDs tend to be less aware of their weaknesses and strengths in self-reports.
RE: Thank you for this suggestion, we added this limitation (lines 416-418).
We hope that we have addressed all of the comments and recommendations and look forward to seeing a revised version of the manuscript ready for publication in Children.
Sincerely,
Authors
Reviewer 2 Report
Comments and Suggestions for Authors
Dear authors,
Please find attached the review of your article,
Yours sincerely

Author Response
Dear Reviewer;
Thank you for your helpful and useful comments to improve the quality of our paper.
We revised our paper according to these comments. Please see below the changes we have made – in the text all corrections/additions are highlighted in red to make it easy to see what specific changes were made.
Reviewer 2
1. The abstract covers all the points of the article. On the other hand, the authors should review the applicative perspective mentioned at the end of the abstract, which is a little generalist. What do the observed correlations bring to research and practice? The results may seem a littlle fatalistic. How can we use them to improve things from an educational point of view, for example?
RE: Thank you for this suggestion, we corrected the conclusion part of the Abstract,
2. Are the percentages given at the start of the introducton global figures? Is this true everywhere in the world? It might be useful to clarify this and add scientific references.
RE: This is a global rate and based on the report of the World Health Organization (Mental Health of Adolescents. 2021. Retrieved 20.09.2023, from https://who.int/news-room/fact-sheets/detail/adolescent-mental-health)
3. The authors say that not all schools are able to provide the necessary conditions of reception and support for pupils in difficulty. Why not? In which country? In France, for example, all schools should be able to provide support for these children, since special arrangements exist and are put in place every year for thousands of children.
RE: Thank you for this note. Unfortunately not all the countries even within the EU are ready to provide the necessary support. See Lines 53-60: “For example, despite the existence of regulations, guidelines, and instructions in Hungary regarding the suitable format and caliber of education for individuals with learning disabilities, not all regions are able to provide the necessary level of integration in education and training. This is primarily due to the inadequate number of special education teachers and the old-fashioned teaching approaches employed by the aging teaching staff.”
4. The authors say that children with learning difficulties are prone to health and socioemotional problems. But the authors justify this by citing two or three adult studies. I don't think this is relevant. It seems to me that numerous studies exist linking these variables in children.
RE: Thank you for this remark, we added some relevant references here (see lines 78-82).
5. Similarly, why do the authors spend time on studies of autisic children? Will autisic children also be part of the research sample?
RE: We mentioned autistic children and those with intellectual disability in relation to proactive coping since there is a lack of studies with children having SLD on this issue. We also added one more reference including students with SLD (lines 92-99).
6. In my opinion, the introduction is poorly organized. The authors should certainly revise the plan to take the reader to the important part. Perhaps it would be wise to have a more canonical plan: recent concept definitions (2023), link between difficulties and cognition in children, link between difficulties and emotion in children, link between difficulties and socialization in children, problematic, hypotheses.
RE: Thank you for this note. Based on your and the other Reviewer’s suggestions, we restructured the Introduction as follows: 1) recent concept and definitions; 2) difficulties among children with SLD; 3) Strengths among children with SLD. In addition, we also mentioned when SDQ was used in a concrete study. Some new references were also added. We really hope that Introduction is now better organized.
7. It would be necessary to include a table detailing the distribution of children. It seems to me that the sample is small for each variable modality. If I understand correctly, there are 276 adolescents, half of them with learning difficulties, i.e. around 140 (the exact figures are not given), and these 140 children with learning difficulties are aged 11 to 18, i.e. 140/7 = 20. There would be around 20 adolescents per age group. Is this correct? Is age an independent variable? If so, how? If not, the authors should justify this choice. Why group together all children aged 11 to 18? This seems to be methodologically unreliable, as it adds a number of spurious developmental variables.
RE: We agree that age may be a relevant variable here. Therefore, we added a table for the sample characteristics as the other Reviewer suggested. In addition, we made two age groups (11-14-year-olds and 15-18-year-olds) and run analyses separately for these categories.
8. What's more, children with learning difficulties have extremely heterogeneous difficulty profiles. How can we make scientifically sound comparisons? Obviously, having access to this population and enough subjects to conduct analyses is very difficult, but grouping children with such intrinsic differences seems inoperable.
RE: In terms of heterogenity, in this sample, 51.8% hade special educational needs due to specific learning disorders. The following description was added (liners 143-153):
“Using this design, in the sample, 51.8% (n = 143) had special educational needs due to specific learning disorders. Of these, 95% of the sample (136 children) have a mixed disorder, with difficulties in all three areas (dyslexia, dysgraphia and dyscalculia) (49 of these have a diagnosis of F83 - Mixed specific developmental disorders and 87 have a diagnosis of F81.3 - Mixed disorder of scholastic skills according to the ICD-10 categorisation. In the sample, 2 persons have dyscalculia only (diagnosed with F81.2 according to ICD-10), 3 persons have dysgraphia only (diagnosed with F81.1 according to ICD-10) and 2 persons have dyslexia and dysgraphia combined (diagnosed with F81.0 and F81.1 according to ICD-10). One case of ADHD and one case of autism are comorbid diagnoses. Due to the relatively low sample size and various diagnoses, subgroups could not be further defined in the analyses.” However, we did not group the children, only the questionnaires.
9. A section on procedure seems to be missing. How exactly were the questionnaires administered? Did the children have to read the questions themselves? This is a problem with some children. How did the authors ensure that the statements were correctly understood?
RE: We added the following note: “A special educator from the school helped the students with administration and described a diagnosis on the form; otherwise the completion was anonymous. During the recording, those who had problems with reading or interpreting the questions were read out by these facilitators.” (lines 158-162).
10. After reading the results, I understand that age is not an independent variable. This is a major problem. I don't think it's possible to compare 10-year-olds and 18-year-olds on emotional, social and well-being issues. Research in developmental psychology is unanimous: we need to rethink the experimental design and add developmental positioning elements.
RE: We understand your opinion and agree with it; although most studies (e.g., WHO, 2019; Grasso et al., 2022; Ayar et al., 2019) investigated chilren with a great age range (e.g., 7-15 or 10-19 years), there may be age-specific diferences in various age groups. Therefore, we run the statistical analyses for a younger (11-14-year-olds) and an older (15-18-year-olds) age group.
We hope that we have addressed all of the comments and recommendations and look forward to seeing a revised version of the manuscript ready for publication in Children.
Sincerely,
Authors
Round 2
Reviewer 1 Report
Comments and Suggestions for Authors
In general, the authors have adequately addressed the issues raised. However, they should revise the numbering of the tables in the text (e.g. Table 1, Table 2, Table 3,...) since not all of them are numbered and some appear twice.
Author Response
Dear Reviewer;
Thank you for useful comment. Please see our reply to it.
Reviewer 1.
In general, the authors have adequately addressed the issues raised. However, they should revise the numbering of the tables in the text (e.g. Table 1, Table 2, Table 3,...) since not all of them are numbered and some appear twice.
RE: Thank you very much for noticing this failure; we are really grateful for it. We have made the necessary changes in numbering the tables.
Sincerely,
Authors
Reviewer 2 Report
Comments and Suggestions for Authors
Thank you for all these clarifications and for making the effort to distinguish 2 age groups to limit developmental bias. Nevertheless, I still feel that the authors must justify the age groups selected from a psychological and developmental point of view. Why this choice with regard to the development of emotional, social and cognitive skills? What theoretical models are they basing themselves on?
I think this is important.
Author Response
Dear Reviewer;
Thank you for your helpful and useful comment to improve the quality of our paper. We have tried to do our best to meet your request.
Reviewer 2.
Thank you for all these clarifications and for making the effort to distinguish 2 age groups to limit developmental bias. Nevertheless, I still feel that the authors must justify the age groups selected from a psychological and developmental point of view. Why this choice with regard to the development of emotional, social and cognitive skills? What theoretical models are they basing themselves on? I think this is important.
RE: Thank you for this note; we agree with you that this choice needs a rationale. Therefore, we have added some more references and theoretical considerations in Introduction (see lines 131-144).
Sincerely,
Authors